# Do Diet and Dietary Supplements Mitigate Clinical Outcomes in COVID-19?

**DOI:** 10.3390/nu14091909

**Published:** 2022-05-02

**Authors:** Bhavdeep Singh, Eli Eshaghian, Judith Chuang, Mihai Covasa

**Affiliations:** 1Department of Basic Medical Sciences, College of Osteopathic Medicine, Western University of Health Sciences, Pomona, CA 91766, USA; bhavdeep.singh@westernu.edu (B.S.); eli.eshaghian@westernu.edu (E.E.); judith.chuang@westernu.edu (J.C.); 2Department of Biomedical Sciences, College of Medicine and Biological Sciences, University of Suceava, 7200229 Suceava, Romania

**Keywords:** COVID-19, nutrition, diet, vitamin, dietary supplement, SARS-CoV-2

## Abstract

The coronavirus disease 2019 (COVID-19) has caused a pandemic and upheaval that health authorities and citizens around the globe are still grappling with to this day. While public health measures, vaccine development, and new therapeutics have made great strides in understanding and managing the pandemic, there has been an increasing focus on the potential roles of diet and supplementation in disease prevention and adjuvant treatment. In the literature, the impact of nutrition on other respiratory illnesses, including the common cold, pneumonia, and influenza, has been widely demonstrated in both animal and human models. However, there is much less research on the impact related to COVID-19. The present study discusses the potential uses of diets, vitamins, and supplements, including the Mediterranean diet, glutathione, zinc, and traditional Chinese medicine, in the prevention of infection and severe illness. The evidence demonstrating the efficacy of diet supplementation on infection risk, disease duration, severity, and recovery is mixed and inconsistent. More clinical trials are necessary in order to clearly demonstrate the contribution of nutrition and to guide potential therapeutic protocols.

## 1. Introduction

COVID-19, caused by the SARS-CoV-2 virus, is a novel infectious disease that has led to a global pandemic. The virus gains cellular entry via the ACE2 receptor localized primarily in the lungs and gut of the human body. Spread through respiratory droplets, its clinical manifestations cover a wide range, including symptoms of fever, chills, cough, shortness of breath, loss of taste or smell, headaches, and muscle aches [1]. Infection with COVID-19 can induce a commanding innate and adaptive immune response, with T-cell-mediated adaptive immune responses being essential for viral clearance. Although this initially serves as a protective role, the exacerbation of inflammatory responses can quickly deteriorate the condition of patients with SARS-CoV-2. Normally, viral infection such as COVID-19 is detected by macrophages or dendritic-presenting cells, and viral-derived peptides are presented to TH0 (CD4 cells). They subdivide to either TH1 cells, which secrete IFN-γ that plays an integral role in elimination of virus, or TH2 cells, which trigger a humoral response [2]. In a healthy immune environment, balance between the T-cell subset is pivotal to viral clearance and return to homeostasis. When dysregulated, a sustained inflammatory state can result in T-cell exhaustion and an increased production of pro-inflammatory cytokines, resulting in increased severity of disease. The increased production of proinflammatory cytokines, such as IL-6, IL-1, and TNF-α, can induce a cascade phenomenon labeled cytokine storm in which hyperactive immune response results in damage to vasculature, capillaries, alveoli in lungs, and systemic organ failure, including acute respiratory syndrome (ARDS), resulting in a high mortality rate [3]. Individuals with associated comorbidities such as obesity, diabetes, or cardiovascular disease who are already in a proinflammatory state have increased susceptibility to the cytokine storm. Therefore, proper nutrition is postulated to contribute to maintaining health and could provide the organism with proper support and preventative measures against COVID-19. The lockdown and quarantine restrictions imposed in most countries to stave off the spread of viral infection has led to significant changes in people’s lives and behavior, ranging from restricted movement and sedentary behavior to accessibility to foods, increased stress, and anxiety that could impact nutritional choices and subsequent effects on the immune system’s ability to fight off infection. In an Italian study with 3533 responders to a web-survey of ages between 12 and 86, of which 76.1% were women, changes in lifestyle and eating habits during the lockdown were assessed. The authors reported that 46.1% of responders declared no changes, while 16.7% felt to have improved and 37.2% felt to have made them worse. No significant changes were reported in the physical activity of those who were previously active. In terms of eating habits, 17.7% of respondents reported decreased appetite, while 34.4% had an increase in their appetite. There was an increase in the home recipes used, legumes, cereals, white meat, and hot beverage consumption, while there was a decrease in fresh fish, packaging sweets and baked products, delivery food, and alcoholic intake [4]. Several reviews have been conducted on the effects of nutrients on COVID 19 (see References [5,6], for example). This review integrates the effects of specific nutrients thought to play a major role in COVID-19 and their mechanisms of actions by examining their role and efficacy in human clinical trials. Furthermore, we evaluated the effects of several widely used diets, such as Mediterranean and Western diets, and their impact on COVID-19 manifestations. Finally, we evaluated studies on the use of traditional Chinese medicine in the prevention and treatment of COVID-19 and the effects of probiotics as food supplements in modulating the gut microbiota and subsequent effects on COVID-19 patients.

## 2. Diet and Its Impact on Risk of Infection

Reduction of inflammation in individuals, irrespective of their underlying conditions, has been proposed to decrease susceptibility to COVID-19 infection and its associated symptoms. Among the numerous diets available that promote healthy outcomes, the Mediterranean diet has been studied the most, as it relates to its effects in the fight against COVID-19. This diet emphasizes fruits, root vegetables, beans, nuts, legumes, seeds, and olive oil and places fish and seafood above poultry, meats, and sweets consumption. The effects of these foods and nutrients provided on overall health and disease prevention is well documented. For example, consumption of fruits and vegetables that constitutes the core of this diet results in an increase in dietary polyphenols that act as antioxidants, blocking pro inflammatory cytokines, suppressing inflammatory gene expression, and reducing oxidative stress, leading to decreased overall inflammation [7]. Likewise, this diet’s rich content in antioxidant vitamins and omega-3 fatty acids derived from fish is thought to protect against oxidation of polyunsaturated fatty acids in cell membranes and reduce levels of proinflammatory mediators, such as tumor necrosis factor alpha. For example, a cross-sectional study involving 432 subjects showed that the Mediterranean diet was associated with an increase in adiponectin, which mediates the body’s response to insulin and has anti-inflammatory effects modulating the pathogenesis of atherosclerosis and other diseases [8]. 

In addition to reduced systemic inflammation and oxidative stress, a large cross-sectional study conducted in Attica, Greece, showed that adherence to the Mediterranean diet has been associated with decreased insulin resistance, increased incretins, low blood pressure; reduced arterial stiffness; and reduced plasma glucose, LDL cholesterol, and triglycerides, all of which provide beneficial effects in several diseases, such as diabetes, hypertension and cardiovascular [9]. Further, the Mediterranean diet decreases levels of C reactive protein, a biomarker for inflammation; and interleukin-6 and interleukin-8, which are involved in proinflammatory response [10]. Most important, a prospective cohort study involving 158 asthmatic children and 50 non-asthmatic children followed for 22 weeks showed that a Mediterranean diet can have a positive impact on symptoms associated with rhinitis, asthma, and overall respiratory function, as well as reduction in oxidative stress in asthmatic children exposed to air pollutants, suggesting a possible protective mechanism of the Mediterranean diet against COVID-19 [11]. Considering that inflammation is the hallmark condition of COVID-19, the Mediterranean diet has been recommended not only for overall good health maintenance but also as preventative measures in the fight against inflammatory conditions, including COVID-19. In one ecological study across 17 regions in Spain with Mediterranean diet adherence, using an unadjusted linear model, there was a negative association between the Mediterranean diet and COVID-19 cases and related deaths. The Mediterranean diet was measured by using the Mediterranean Adequacy Index score, which was calculated as a quotient, dividing the energy provided by typical Mediterranean food items by the energy provided from non-Mediterranean food items. A higher score indicated a greater adherence to the Mediterranean diet. Using a multivariable linear model to adjust for social determinants of health (income, education, housing, environment, and life satisfaction), a stronger negative association between Mediterranean Adequacy Index score and COVID-19 cases and COVID-19 related deaths was found. Of note, these findings pre-dated the approval of medical treatments known to reduce mortality of COVID-19, such as dexamethasone and remdesivir. This study further examined 23 countries reporting that, when controlling for well-being factors (income, education, housing, environment, and life satisfaction), there was a significant negative association between Mediterranean Adequacy Index score and COVID-19 deaths. These data were obtained from the Organization of Economic Cooperation and Development in which Mediterranean Adequacy Index score per country was reported. There was no control group in this study, and the specific diet was not measured [12]. Although many of these studies do not include a standardized treatment and are not controlled and more work is needed to establish a causative association, preliminary research on the Mediterranean diet is promising. 

In contrast to the Mediterranean diet, the Western diet has been associated with increased inflammatory processes contributing to increased susceptibility to diseases such as CVD, diabetes, and COVID-19 [13]. The Western diet consists mainly of sugar, refined carbohydrates, and saturated fats. It is well documented that consumption of saturated fatty acids can activate macrophages, dendritic cells, and neutrophils through Toll-like receptor 4 (TLR-4) via a proinflammatory cascade pathway, leading to chronic inflammation and impairment of the innate immune response over time [14]. Additionally, consumption of this diet weakens the adaptive immunity via its impairment of T- and B-cell proliferation and induction of B-cell apoptosis through oxidative stress [15]. The consequence of this is a weakened innate and adaptive immune system contributing to increased vulnerability to infections. The reduction in the body’s defense system can lead to systemic diseases affecting various organs, including the lungs. Interestingly, the high fat intake characteristic of the Western diet has been linked with increased macrophages in the lungs, promoting airway inflammation. Furthermore, the Western diet impairs the adaptive immunity, while increasing the innate immunity, thus contributing to chronic inflammation and resulting in increased susceptibility to viral pathogens [16]. Obesity, which is linked to the Western diet, results in increases in Th1 cells and cytotoxic T cells within adipose tissue, while decreasing Th2 cells subset and Tregs cells, which both contribute to reducing inflammation [17]. Increases in Th1 subset absent regulatory processes contribute to a proinflammatory state with impaired immunity (Figure 1). Given that patients with COVID-19 have a high rate of lung infection, such as pneumonia, a poor Westernized diet has been considered one significant culprit contributing to the high severity of associated symptoms [12]. Taken together, the evidence thus far suggests that diets rich in fruit, vegetables, and phytochemicals with anti-inflammatory and antioxidant properties and low in saturated fats may have a protective effect on lung health and may be used as an additional tool in mitigating COVID-19 symptoms [11]. 

As such, a number of different nutritional supplements were hypothesized to serve as therapeutics against COVID-19. Supplements rich in antioxidants and anti-inflammatory properties have been proposed to not only serve as protective agents against COVID-19 infection but also help alleviate symptoms of COVID-19 and decrease morbidity and mortality. This review examines the evidence related to the role nutritional supplementations play in COVID-19 disease. Current studies and clinical trials examining the role of glutathione, zinc, various vitamins, traditional Chinese medicine, and probiotics and their mechanisms of action on COVID-19 pathogenesis were evaluated. The summary of their effects is presented in Table 1.

## 3. Vitamin A

Retinoids are a family derived from vitamin A that plays a role in the immune system. Retinoids stimulate natural killer cells, dendritic cells, and T cells that actively help fight off infections and are at the core of our innate and adaptive immunity [55]. Vitamin A is considered a key regulator in innate and adaptive immune cells and can promote immune homeostasis [56]. Previous studies examining vitamin A’s role in virus infections, particularly enterovirus 71, have shown that all trans retinoic acid (derivative of vitamin A) reduces the percentage of enterovirus 71 (EV71)-infected cells and contributes to reduction in EV71-induced apoptosis [57]. Natural and synthetic retinoids also have direct inhibitory effects on replication of a number of viruses, including hepatitis B virus (HBV), cytomegalovirus, influenza, MeV, and norovirus [55]. Several studies have linked vitamin A to faster recovery from certain viral infections. For example, in a double-blind randomized study involving 189 children diagnosed with measles, it was found that patients who received vitamin A as a supplement were more likely to recover from pneumonia and diarrhea faster, had a shorter hospital stay, and had a reduced risk of death [58]. Very few studies examined the association between vitamin A with COVID-19. In one study examining vitamin A levels in COVID-19 patients, it was found that reduced vitamin A levels correlated with increased inflammatory markers. Patients who were critically ill had significantly lower vitamin A levels than those who were moderately ill with severe reduction in vitamin A, contributing to higher chances of development of acute respiratory distress [19]. Currently, there is an ongoing trial examining 13-Cis-Retinoic Acid as a therapeutical option for COVID-19 [18]. Considering the role of vitamin A in immune functions, its high safety prolife, with side effects of nausea, vomiting, and headache associated with high toxic levels, and previous association with other viral infections, its use in COVID-19 may warrant further investigation [56]. Furthermore, given its known teratogenic effects, its usage requires careful consideration in pregnant women and women attempting pregnancy [59]. 

## 4. Vitamin D

Vitamin D plays a role in both the innate and adaptive systems by acting as an immunomodulator. Briefly, the active form of vitamin D, 1,25(OH)_2_D, increases production of defensin β2 and cathelicidin antimicrobial peptide (CAMP) in monocytes and neutrophils that function to disrupt pathogen cell membrane [60]. In vitro studies have shown vitamin D’s ability to inhibit TH1 cells, reducing production of proinflammatory cytokines such as IFN-γ, IL-6, IL-2, and TNF-α. Although more controversial, 1,25(OH)_2_D has been linked to promote Th2 cell development and production of cytokines: IL4, IL-5, and IL-10 [61,62]. Vitamin D additionally inhibits proinflammatory cells such as TH17 cells and promotes the formation of Tregs cells, which prevent the body from entering the pro-inflammatory stage, serving as regulators of the immune system [63,64]. High Tregs cells are inversely related with respiratory viral disease [65]. Interestingly, recent evidence suggests that vitamin D’s effects can occur seasonally. In a study analyzing 454 peripheral blood mononuclear cell samples from 109 children, it was shown that vitamin D receptor expression was upregulated in the summer months [66]. These findings corroborate the results from a recent study involving 977 volunteer male late adolescents. When controlling for body fat, time spent gaming/TV-watching, smoking, recreational physical activity, and hemoglobin, which are all associated either negatively or positively with physical fitness performance, vitamin D levels modulated physical performance in interaction with seasonality. In the seasons with the mildest climate, the higher level of vitamin D had higher scores of physical fitness [67]. Therefore, high vitamin D levels leading to increased physical fitness can be hypothesized to contribute to improve immunity either pre-infection or during infection. 

Previous supplementation of vitamin D in viral infection has shown promising results, and it has been proposed for usage against COVID-19. In one double-blind study with 167 patients, supplementation of vitamin D helped reduce the incidence of seasonal influenza possibly due to increased production of antimicrobials and reduction of proinflammatory cytokines [67]. Additionally, a meta-analysis examining 25 randomized controlled trials revealed that vitamin D supplementation helped reduce the risk of acute respiratory infection and provided a protective value when given daily or weekly, without any additional bolus. Supplementation also reduced the rate of asthma exacerbation requiring treatment [20].

Several studies have examined the links between vitamin D and COVID-19 since the pandemic began, given its role in the immune system. In one study examining 235 patients hospitalized with COVID-19, a relative risk of 1.59 was associated with severity of COVID-19 in patients with vitamin D levels less than 30 ng/mL. The cross-sectional design, as well as a lack of COVID tests for all patients with COVID-19 symptoms, however, did limit the significance of the results [20]. This study was supported by a retrospective observational study conducted on 185 patients that found that vitamin D deficiency was associated with higher risk of invasive mechanical disease and/or death [21]. In another study, Meltzer et al. categorized 489 patients hospitalized into groups based on the status of their vitamin D reported in their medical chart within 1 year of being treated for COVID-19. They found that patients with likely low levels of vitamin D had an increased likelihood of testing positive for COVID-19 compared to those individuals who had sufficient vitamin D levels [22], suggesting a possible protective role provided by adequate vitamin D levels. These findings are in agreement with those of Kaufman et al., who found that patients with less than 20 nm/mL 25(OH)D values had a higher COVID-19 positivity rate compared to patients with normal vitamin D levels in a study involving 191,779 patients in the United States. This relationship remained true after adjustment of confounding factors such as ethnicity, sex location, and age [23]. It is important to note that not all studies have shown an association between vitamin D deficiency and COVID-19. For example, examination of the UK biobank by Hastie et al. found no association between vitamin D and COVID-19 infection after adjustment for confounding variables [24]. Several studies examined whether lockdown conditions had an impact on vitamin D levels. When adolescent males aged 18–19 enrolled in the army in Southern Switzerland (*n* = 298), it was found that 25-hydroxyvitamin D levels were not different in 2014–2016 as compared with the 2020 study period. Furthermore, the levels were similar across all months. The levels of vitamin D were measured between July and December 2020 [68]. To examine changes in potential vitamin D deficiency pre- and post-outbreak, 303 infants and toddlers aged 2–24 months were sampled for 25-hydroxyvitamin D from 1 June 2019 through 30 November 2020. June to November 2019 was considered “pre-outbreak”, and June 2020 to November 2020 was considered “post-outbreak”. The study was also stratified by age. There were no immediate changes in serum 25-hydroxyvitamin D after the start of COVID-19, but in the subsequent months, it continued to decline. This decreasing trend reached significance for the young infants (2–7 months) after adjusting for age, sex, and UV index. Significance was never met for the older infants (7–24 months) [69]. Although a reduction in vitamin D levels may have reduced in infants, the same was not true for older males in Switzerland. The evidence is inconclusive and, as a result, is not reliable to use when considering the effect of vitamin D deficiency on COVID-19. Interestingly a recent meta-analysis showed vitamin D playing a protective role in adults against respiratory tract infections but not in other regions of the world. It may be possible that similar effects may be seen in other countries and the results could have been due to the lack of few European studies [70].

Examination of vitamin D’s role in COVID-19 experimental trials have yielded interesting results. To determine whether bolus vitamin D supplementation could improve survival rate, Annweiler et al. placed 77 patients hospitalized for COVID-19 into three different groups: those who had regularly supplemented with vitamin D (50,000 IU vitamin D3 per month, or 80,000 IU or 100,000 IU vitamin D3 every 2–3 months) over the past year, those who were given vitamin D after COVID-19 diagnosis (oral supplement of 80,000 IU vitamin D3 within a few hours of COVID-19 diagnosis) and the control group who received no vitamin D supplement. When examining 14-day mortality, they found that patients who had vitamin D supplementation in the past year prior to COVID-19 had longer survival time. There was no significant difference between patients given vitamin D after their diagnosis and the control group leading to the conclusion that regular vitamin D supplementation was associated with better survivability in COVID-19 patients [25]. In a quasi-experimental study involving 66 residents in a nursing home, bolus vitamin D supplementation during COVID-19 or in the preceding month was associated with less severe COVID-19 symptoms and better survival rate [71]. Likewise, individuals who received supplementation of vitamin D, magnesium, and vitamin B12 (DMB) were less likely to require oxygen therapy or need intensive care support compared to the control group. However, the combination of various supplementation along with the small sample size of 43 patients does limit the significance of these results [26]. Murai et al. investigated whether a single high dose of vitamin D could have an effect on hospital length of stay in COVID-19 patients. Using a double-blind randomized placebo-controlled study with 240 hospitalized patients with COVID-19—120 of which received 200,000 IU vitamin D3 and 120 of which did not—there were no significant difference in hospital length of stay, in hospital mortality, need for mechanical ventilation or intensive care unit between the group receiving the vitamin D supplementation and the control. This high dose was well tolerated by patients with just one adverse event of vomiting following administration [27]. Lastly, a recent combined retrospective and prospective study was conducted to evaluate vitamin D supplementation in the treatment of COVID-19. This study concluded that vitamin D treatment shortened the hospital stay in COVID-19 patients even with the existence of comorbid conditions in addition to decreasing mortality rate by 2.14 times [28]. 

Taken together, studies examining vitamin D and COVID-19 show some association between individuals who have adequate levels of vitamin D prior to infection and COVID-19 disease severity. However, further investigation is warranted to determine whether supplementation of vitamin D can be used as a therapeutic agent against COVID-19. It may be possible that vitamin D may play more of a preventive role against COVID-19 rather than an immediate therapeutic effect in individuals infected. Given its toleration and low risk of toxicity, which includes nausea and vomiting secondary to hyperkalemia, and its good safety profile, it warrants further examination [72]. There are no current guidelines for dosage of vitamin D for COVID-19 patients, but recent studies suggest that increase in vitamin D levels to 80–100 ng/mL could reduce inflammatory markers, without significant adverse effects, associated with COVID-19 [73]. In addition, new studies have revealed that seasonal factors might need to be considered in the use of vitamin D to treat or prevent COVID-19. However, more controlled studies and randomized trials are needed to further tease out the role vitamin D may play against COVID-19 infection. 

## 5. Vitamins C and E

Antioxidants protect against free radicals, which have been implicated in a variety of diseases, due to their ability to harm cells. One form of these free radicals is reactive oxygen species (ROS) that can be generated from membrane-bound NADPH oxidases (Nox). In the innate immune system, the ROS is vital for assisting in killing microorganisms and protecting the body from pathogens. However, Nox and the oxidative state have implications on susceptibility to diseases, as upregulation of Nox2 has been associated with atherosclerotic plaque [74]. Furthermore, Nox2 has been showcased to suppress antiviral and humoral signaling networks by single stranded RNA and DNA viruses, and its inhibition can contribute to nullifying the pathogenicity of influenza [75].

Increased oxidative stress due to the formation of free radicals from the upregulation of Nox2 in cardiovascular-related complications suggest its possible role in myocardial damage [76,77]. Its activation and production of reactive oxygen species lead to a worse prognosis of patients with cardiovascular issues and diagnosed with COVID-19 [77]. This high inflammatory state weakens the immune system causing more severe symptoms with infection. Antioxidant vitamins, mainly vitamins C and E, could play some role by reducing free radicals. In studies involving administration of antioxidant vitamins, vitamins E and C significantly decreased tissue malondialdehyde levels which are markers of oxidative stress that are produced due to ROS targeting lipids of cell membrane and associated with overall tissue damage [78]. The decrease in ROS resulted in protection of the cell membranes. 

Furthermore, vitamin C has been shown to reduce levels of proinflammatory cytokines, such as TNF-α, which has been implicated in the entry of SARS-CoV-2. Not only can the increased intake of vitamin C promote the production of IL-10, which serves to control inflammation that further propagate the adverse events associated with COVID-19, but it also further strengthens the function of the lung epithelial barrier that regulates alveolar fluid clearance, providing further protection from any lung injury [79,80]. 

Several studies examining whether vitamin C was beneficial in COVID-19 showed conflicting results. For example, in a Chinese study, a high intravenous dose of Vitamin C given to 56 critically ill COVID-19 patients had no effect on invasive mechanical ventilation-free days [33]. Similarly, a retrospective study conducted in Brooklyn on 102 ICU COVID-19 patients who received vitamin C and zinc supplementation showed that there was no change in survival compared to those without the treatment [34]. A randomized controlled trial conducted in Isfahan, Iran, on 72 COVID-19 patients found that supplementation with vitamin C and E had no effect compared with control patients [81]. Likewise, in an open-label, randomized, and controlled trial of 60 patients, where the case group received lopinavir/ritonavir and hydroxychloroquine, along with 6 g of vitamin C, there was no significant difference in oxygen levels, length of intensive care unit (ICU) stay, or mortality compared to the control group receiving just lopinavir/ritonavir and hydroxychloroquine [29]. Furthermore, a meta-analysis study involving six randomized controlled trials and a total of 572 patients found no significant changes in mortality, ICU length of stay, hospital length of stay, or need for invasive mechanical ventilation with vitamin C treatment, irrespective of severity of illness or route of administration [30]. On the other hand, a study conducted at the Michigan hospital on 79 COVID-19 patients who received vitamin C supplementation showed that vitamin C significantly reduced mortality [35]. Furthermore, a recent double-blind randomized clinical trial showed a higher mean survival duration in patients following a 14-day supplementation of 500 mg of vitamin C. A reduction in potassium levels was also found; however, serum potassium levels were within the normal range for both groups, and, thus, the effect was not clinically significant. The study was also limited in regard to data on length of hospital stay and ventilator-free days [31]. Similarly, a single center randomized clinical trial involving 54 patients reported an improvement in oxygen saturation, a reduction in respiratory rate, and a reduction in rate of lung involvement in the treatment group receiving 2 g of vitamin C every 6 h for 5 days compared to the control group [32]. Together, these studies show inconsistent results, with some data showing significant improvement, whereas others do not show any effects with supplementation or high doses of vitamin C [36]. In these studies, vitamin C doses were generally tolerated well, with increasing vitamin C being associated with increased oxalate excretion with any significant clinical adverse effect, but further research must be conducted to determine the recommended dose in patients with COVID-19. Previous research has shown safe doses of vitamin C being less than 1000 mg daily, but additional studies must be conducted regarding vitamin C dosage and its safety prolife in patients with COVID-19 [82,83]. 

Few studies examined the role of vitamin E in COVID-19. A meta-analysis study with 135,967 individuals from over 19 different clinical trials showed that a high dosage (>400 IU/day) of vitamin E supplementation may increase all-cause mortality. This risk was increased with increased dosage [84]. Studies examining vitamin E supplementation are scarce. Considering the role of vitamin C as an antioxidant and its promising effects in reducing the length of time of other diseases, more studies with increased power should be conducted in order to better understand whether vitamins C and E are making a significant difference in improving the outcomes of patients with COVID-19. 

## 6. Glutathione

Glutathione (GSH) is an antioxidant present in plant and animal cells that may be used to treat COVID-19 symptoms of dyspnea, pneumonia, and acute respiratory distress syndrome. It plays an important role in antioxidant defense by reducing reactive oxygen species and regulation of cellular processes, including gene expression, DNA and protein synthesis, and cell proliferation and apoptosis [85]. Several studies have found that glutathione can inhibit NF-κB activation and signaling that affect inflammation in disease processes [86]. It is also found in abundance in the lining fluid of the lungs to protect epithelial cells and has been shown to be depleted in patients with cystic fibrosis and smokers, indicating its protective role in the respiratory tract [87]. GSH deficiency has been associated with greater susceptibility to viral infections, greater symptom severity, and poor prognosis [88].

N-acetylcysteine (NAC), a GSH precursor required for its synthesis, has been shown to block NF-κB and cytokine formation and restore or protect against glutathione depletion caused by disease processes [89]. It has a dose-dependent effect on respiratory conditions and viral infections, including chronic obstructive pulmonary disease, H1N1 influenza, and respiratory syncytial virus [90,91,92]. In a small, controlled study measuring glutathione levels in patients with COVID-19 versus those without with 29 total study subjects (21 with COVID-19, 8 without), it was found that those with COVID-19 had a 60% lower RBC concentration of total-GSH and reduced-GSH [93]. Another study on COVID-19′s relationship with GSH enrolled 96 patients: 19 in the intensive-care unit with endotracheal intubation, 35 hospitalized not in the intensive-care unit, 24 in the intensive-care unit without endotracheal intubation, and 18 healthy individuals. Glutathione was measured and showed that levels of GSH had an indirect relationship with fever and duration of hospitalization and a direct relationship with peripheral oxygen saturation SpO2 [94]. In a larger meta-analysis study involving a total of five randomized controlled trials and 183 patients, NAC reduced the duration of intensive-care-unit stay in patients with ARDS but did not significantly reduce short-term mortality risk or 30-day mortality [95]). Furthermore, in a randomized clinical trial involving 40 mechanically ventilated patients, those in the NAC group saw a statistically significant increase in oxygen saturation compared to the control. There was no significant improvement in mucous secretion density compared to the control group, and this might have been explained by the reduced NAC dosage of 2 mL of NAC in 8 mL of normal saline. The reduction of the dosage was performed to avoid adverse effects of bronchospasm, fever, or rhinorrhea [96]. When it comes to the recommended dose, further research must be performed to determine the therapeutic dosage for patients impacted by COVID-19. Generally, NAC has been shown to be safe in clinical trials, with the side effects of oral NAC including nausea, vomiting, diarrhea, flatus, and gastroesophageal reflux, as well as IV NAC causing anaphylactoid reactions [97]. Currently, there are multiple trials examining NAC as a treatment option. A recent randomized clinical trial led by the Baylor College of Medicine has been assessing the response to supplementation of glutathione precursors glycine and cysteine on oxidative stress, inflammation, and endothelial dysfunction in patients with COVID-19 [37]. Additionally, a nonrandomized clinical trial involving 48 patients is underway to determine if NAC can clinically improve patients with COVID-19 [38]. This is a promising and necessary study to further understand the potential use of GSH and NAC in COVID-19 treatment. Given the limited data available, further clinical trials with a larger sample need to be conducted to determine the role that glutathione can play in therapeutical treatments for COVID-19. 

## 7. Zinc

Micronutrients and trace elements such as zinc have been receiving more attention for their immunoregulatory and antiviral properties and their potential use in combating COVID-19. Among trace elements, zinc has garnered much interest for its essential role as a cofactor, signaling molecule, and structural element in numerous cell growth and development, and DNA and protein synthesis processes [98,99]. Zinc is also extensively involved in both innate and acquired immune responses. It may improve host defense by maintaining the structural proteins β-catenin and E-cadherin in the respiratory epithelium barrier, while zinc deficiency has been shown to compromise barrier function by upregulating IFNγ and TNFα, enhancing FasR signaling and leading, thus, to apoptosis [100]. This can contribute to the increased susceptibility to viral respiratory infections. Zinc is critical in the fast growth, differentiation, and activation of immune cells [101]. Not only does this increase immune cell proliferation, but it can also enhance cellular resistance to apoptosis by inhibiting caspases and increasing the Bcl-2/Bax ratio [102]. Zinc has immunomodulatory properties related to preventing cytokine overproduction, which can be essential in avoiding hyperimmune responses [103,104]. Zinc can also directly inhibit viral replication by preventing fusion with the host membrane and impairing viral polymerase function and protein translation and processing [105]. The wide-ranging antiviral and immunomodulatory functions highlight the value of zinc in maintaining health, preventing disease, and developing treatments.

While there is growing interest in assessing the preventive and therapeutic potential of zinc against SARS-CoV-2, its direct effects in improving COVID-19 are still unclear. Several observational studies have assessed the association between serum zinc levels in relation to malnutrition with COVID-19 morbidity and mortality. An observational study of 269 patients demonstrated an association between low plasma zinc levels with severe acute respiratory distress syndrome [39]. Another observational study of 249 patients found a significant negative correlation between serum zinc levels and inflammation, as measured by C-reactive protein and IL-6 [40]. Zinc may hold potential as part of supplemental therapy for COVID-19, as it may enhance the efficacy of other treatments. A more recent clinical trial using 214 outpatient patients examining the effects of zinc gluconate alone, vitamin C alone, or combined zinc and vitamin C on the number of days until 50% reduction of symptoms was performed. It took 5.5 days for the vitamin C and combined treatment group to reach 50% reduction of symptoms, whereas the zinc group required 5.9 days compared to 6.7 days for the controls. None of these differences were considered statistically significant, however [44]. An observational retrospective analysis of data from patients hospitalized with COVID-19 was conducted to test whether adding zinc to hydroxychloroquine and azithromycin would make a difference in hospital outcomes. In this study, 411 patients took zinc sulfate in addition to the hydroxychloroquine and azithromycin, and 521 patients did not take the additional zinc. The dosage of zinc was a 220 mg capsule with 50 mg of elemental zinc twice daily for 5 days. In univariate analysis, zinc was found to not make any difference on length of hospital stay, duration of mechanical ventilation, maximum oxygen flow rate, average oxygen flow rate, average fraction of inspired oxygen, or maximum fraction of inspired oxygen. What was found to be significant in this study was that zinc sulfate added to hydroxychloroquine and azithromycin was associated with a decrease in mortality or transition to hospice among patients who did not require ICU level of care [106]. Other clinical trials are currently recruiting members to test whether zinc can be an effective symptomatic treatment for COVID-19 [41,42,43]. 

Several studies have examined the effects of zinc prophylaxis and therapy in other respiratory and viral illnesses, including dengue virus infection, malaria, measles, tuberculosis, pneumonia, and, most notably, the common cold, with conflicting results. [107,108]. For example, a 2012 meta-analysis evaluated 17 randomized controlled trials with 2121 total participants and found that oral zinc formulations shortened the duration of common cold symptoms in adults [109]. However, while these effects were supported by another meta-analysis of 22 studies conducted in 2020, it also showed that zinc supplementation did not reduce symptom severity [110]. Both meta-analyses cited limitations such as heterogeneity in dosage and formulation, low external validity, and self-reported symptoms. Interestingly, zinc is effective in inhibiting RNA-dependent RNA polymerase activity in vitro, a core enzyme in the replication of positive-stranded RNA viruses such as coronavirus [111]. Interestingly, a recent meta-analysis conducted on nutritional supplementation for prevention of vital respiratory tract infections (RTIs) found that zinc supplementation had a protective effect against RTI in children from Asia. However, the results could have been explained by initial nutritional status, and heterogenous effect sizes was cited as a limitation of the study [70]. Overall, these studies indicate a need for continued research to better understand the potential uses of zinc against COVID-19. 

The therapeutic dosage of zinc is yet to be determined and may prove difficult given a patient’s underlying disease, initial zinc status, and other nutritional requirements. In accordance with the National Institutes of Health, the current guidelines for upper limit for daily zinc intake in an adult is 40 mg [112]. Zinc dosage in patients requiring parenteral nutrition has widely ranged from 3 to 22 mg zinc/day, without side effects [113]. A recent randomized controlled trial using a zinc concentration of 0.24 mg/kg/day, with doses ranging from 12 to 30.7 mg in hospitalized COVID-19 patients, reported no serious adverse events among 94 administrations of high-dose IV zinc [114]. Given the current literature on high-dose zinc supplementation, further studies must be conducted to determine the correct dosage.

## 8. Omega-3 Fatty Acids

Fatty acids are an integral component of cell membrane phospholipids with various structural, signaling, and immune modulating roles [115] Fatty acids can be classified based on the number of bonds the molecule contains as saturated or unsaturated fatty acids. Omega-3 fatty acids, part of the family of polyunsaturated fatty acids, have been shown to act as anti-inflammatory mediators. Omega-3 fatty acids act to reduce inflammation through downregulation of the NF-κB pathway [116]. Two forms of omega-3 fatty acids, eicosapentaenoic acid (EPA) and docosahexaenoic acid (DHA), form resolvins and protectins that exert significant anti-inflammatory effects by inhibiting the synthesis of proinflammatory cytokines such as IL-1 and IL-6, reducing neutrophil trans endothelial migration, and promoting increased macrophage phagocytosis to resolve inflammation [116,117,118]. Reduction in the inflammatory pathway has contributed to the supplement usage of omega-3 fatty acids against cardiovascular disease and rheumatoid arthritis (RA). Extensive research has showcased the inverse relationship between consumption of omega-3 fatty acids and cardiovascular disease; however, this is not further explored here [119,120,121]. In patients with RA, omega-3 supplements resulted in decreased production of IL-1, resulting in an improvement in clinical status [122,123]. Given the reduction in inflammation mediated by omega-3 fatty acids, its supplementation could play a therapeutic or preventative role against COVID-19. In a recent double-blind randomized clinical trial involving 128 critically ill patients with COVID-19, the administration of omega-3 fatty acid resulted in significant improvement of kidney function indicators, arterial pH, and bicarbonate and had a higher one-month survival rate [45]. However, the results of this study should be interpreted with caution, due to the small sample size and the risk for bias in the presentation of results. Although the authors reported that only six (21%) patients in the intervention group survived at least one month compared to two (3%) patients in the control group, the difference of four patients is too small to draw meaningful conclusions. Furthermore, while there was a better 1-month survival rate in the intervention group, the mortality at the end of the 2-week treatment period was higher in the intervention group compared to the control [45]. Finally, there were several limitations of this study, including a lack of information about dietary and caloric intake of the two groups, the status of omega-3 fatty acids at baseline, the drugs used in the intervention group, and whether there were any changes in inflammatory markers between groups [124]. 

In a meta-analysis of three randomized controlled studies involving over 411 mechanically ventilated patients with acute respiratory distress syndrome, a diet with increased omega-3 fatty acids was associated with a significant reduction in the risk of developing new organ failures, time on ventilation, and overall risk of mortality [125]. Further studies are currently underway examining the effects of omega-3 fatty acids against COVID-19 through double-blinded randomized trials to determine if this modality can serve as a treatment option [126]. 

## 9. Traditional Chinese Medicine

Herbal treatments as a form of traditional Chinese medicine (TCM) have long been used to manage illnesses and outbreaks, including the previous coronavirus epidemics of SARS-CoA in 2003, H1N1 influenza in 2009, and MERS-CoV in 2012 [127,128,129]. Various studies conducted on TCM have demonstrated its antiviral, antibacterial, and anti-inflammatory properties that may be beneficial in preventing and treating disease. A meta-analysis of eight randomized controlled trials found that a combined therapy of Chinese herbal medicine (CHM) and conventional Western medicine shortened the duration of fever, reduced chest radiograph abnormalities, and provided symptom relief in patients with SARS [130]. Another meta-analysis found that the H1N1 influenza infection rate was significantly lower in patients who were administered herbal treatments compared to those in the control group, who either received a placebo or did not receive any treatments [128]. These studies indicate the potential clinical benefits of incorporating TCM when treating patients.

Since the onset of the COVID-19 pandemic, many Chinese provinces have developed prevention and control programs that include guidelines on herbal treatment formulations and treatments [131]. This has spurred increased research into the efficacy, safety, and mechanisms of these treatments. While there are a wide variety of herbal recipes available, there are six decoctions and formulations that have been shown to be the most effective in treating patients with COVID-19 [132]. These include Jinhua Qinggan (JHQG) granules, Lianhua Qingwen capsules, lung cleansing and detoxifying decoction, Xuanfeibaidu (XFBD) granules, Huashibaidu (HSBD) granules, and Xuebijing injection [131]. Many of the recipes utilize a wide variety of medicinal herbs, with some herbs used more often than others. When examined the frequency of specific herbs used, it was found that Armenicae Semen Amarum and Ephedrae Herba were among the most frequently used to treat mild, moderate, and severe disease stages, while Glycyrrhizae Radix et Rhizoma was one of the herbs with the highest usage frequency in all disease stages, as well as recovery [133]. 

Jinhua Qinggan (JHQG) was first developed during the H1N1 pandemic and has since been adapted to treat COVID-19. It is composed of Flos Lonicerae, Herba Ephedra Sinica), Gypsum Fibrosum, Semen Armeniacae Amarum, Scutellariae Baicalensis, Fructus Forsythiae Suspensae, Bulbus Fritillariae Thunbergii, Rhizoma Anemarrhenae, Fructus Arctii, Herba Artemisiae Annuae, Herba Menthae Haplocalycis, and Radix Glycyrrhizae [46]. In several double-blinded randomized controlled trials of patients with influenza, a 5-day treatment of JHQG speeded up recovery and significantly reduced serum C-reactive protein and cytokines such as IFN-γ [134,135]. This may suggest its potential in COVID-19 treatment, and some studies have demonstrated promising results. In a study involving 80 patients of which 44 took JHQG granules within 24 h of admission, it was found that the treatment group had shorter viral nucleic acid detection and faster pneumonia recovery time seen through chest CT compared to the control group [46]. 

Due to its broad antiviral, anti-inflammatory, and antioxidant properties, Lianhua Qingwen (LHQW) has been previously used to treat SARS and influenza [136,137]. It is composed of 13 herbs, with the main ingredients including *Lonicera Japonica* (honeysuckle), *Forsythia suspensa*, *Rhodiola rosea*, and *Rheum palmatum*. *Lonicera japonica* and *Forsythia suspensa* may prevent the binding of SARS-CoV2 with the angiotensin converting enzyme, thus inhibiting viral infection [138]. *Rhodiola rosea* possesses antioxidant properties that can reduce pulmonary inflammation by suppressing oxidative stress and apoptosis [139]. *Rheum palmatum* behaves as an anti-inflammatory substance by inhibiting cytokine release [140]. In vitro studies of LHQW revealed significant inhibition of viral replication and reduced cytokine production, including TNF-a and IL-6 [141]. A randomized controlled trial involving 14 days of treatment with LHQW capsules for COVID-19 infection found that time of recovery from fever, fatigue, and coughing was shorter compared to the control group, and there was also a higher recovery rate [47]. The ability of LHQW to shorten the recovery time is helpful, particularly given the lack of treatment options. 

Lung cleansing and detoxifying decoction (LCDD), also known as Qingfei Paidu (QFPD), is a general prescription that consists of 21 herbs that has primarily been used to manage lung infection [137,142]. Polysaccharides are generally thought to be one of the main active ingredients responsible for a wide range of immunomodulatory and anti-inflammatory effects [143]. For example, the *Glycyrrhiza* polysaccharide can promote the maturation, differentiation, and production of lymphocytes and macrophages [144], while the *Polyporus umbellatus* polysaccharide is involved in activating the TLR-4 signaling pathway to stimulate B cells, macrophages, and dendritic cells [145]. Some anti-inflammatory polysaccharides in LCDD can suppress the cytokine storm by targeting the AKT1, MAPK1, MAPK14, IL-6, and TNF pathways [137]. Numerous studies have demonstrated the efficacy of LCDD in treating patients with COVID-19. In 214 confirmed cases, administration of four treatment courses of this medicine resulted in more than 60% of patients showing improvements in their symptoms and the remaining 30% showing stable condition [143]. Another study retrospectively assessed 63 patients with mild-to-moderate disease severity and found that the combination of LCDD with Western medicine significantly reduced inflammatory markers compared to treatment with Western medicine alone [48]. Due to these properties, LCDD has shown to be an efficacious treatment, particularly for mild cases. 

Xuebijing injection (XBJ) is a medicine consisting of five traditional herbs, namely Chishao (Radix Paeoniae Rubra), Danggui (Radix Angelica Sinensis), Chuanxiong (Rhizoma Chuanxiong), Honghua (Flos Carthami), and Danshen (Radix Salviae Miltiorrhizae) [146]. Due to its extensive use to treat bacterial pneumonia, sepsis, and acute respiratory distress syndrome, XBJ has been recommended to treat severe and critical cases of COVID-19 [131]. This intravenous preparation can improve microcirculation, regulate coagulation, and remove toxins [147,148]. It is believed that Xuebijing’s mechanism of action involves the regulation of cytokine production, resulting in the inhibition of pro-inflammatory cytokines [149]. A double-blinded randomized controlled trial of 60 patients found significantly suppressed secretion of IL-6, IL-8, and TNF-alpha and higher levels of lymphocytes after 14 days of XBJ treatment [49]. In a study involving 60 patients with severe stage of COVID-19, 100 mL of XBJ injection resulted in increased white blood cell (WBC) count and decrease in pro-inflammatory markers, such as C reactive protein (CRP) and erythrocyte sedimentation rate (ESR). The increase in WBC and decrease in CRP and ESR were more significant compared to individuals who received only the 50 mL injection, as well as the control [50].

Taken together, these studies showed that, compared to Western medicine alone, integration of TCM with Eastern medicine has been shown to reduce not only the duration of symptoms but improve the rate of clearance. Further studies need to be conducted to determine the validity of these results. However, given that TCM is tolerated well, with minimal side effects, including nausea, vomiting, and possible rash, its integration with Western medicine may be warranted in COVID-19 patients [51,142]. 

## 10. Probiotics

Probiotics such as *Lactobacillus* and *Bifidobacterium* are live microorganisms that have long been consumed as part of a regular diet through dairy and non-dairy fermented products or supplements for its beneficial effects on gut flora [150]. They provide health benefits through a wide range of mechanisms, including activation of the immune reaction by interleukins and natural killer cells. They also induce increased mucosal protection through increased IgA production and the differentiation of CD8+ T cells into cytotoxic T cells and CD4+ T cells into Th1 and Th2 cells [151]. While probiotics are primarily associated with gastrointestinal health, some studies have found that diversity in gut microflora is associated with respiratory diseases, indicating crosstalk between the intestinal tract and lungs via the gut–lung axis to modulate immune responses [152]. Three meta-analyses that assessed a total of 57 randomized controlled trials found that probiotic use decreased the incidence, severity, and duration of respiratory tract infections [153,154].

The current literature examining the potential role of probiotics in COVD-19 prophylaxis and treatment is primarily based on their use for other upper respiratory infections and observations of gastrointestinal dysbiosis in patients with COVID-19. For example, both *Lactobacillus* and *Bifidobacterium* have been shown to improve immune control against influenza in mice [155,156]. Microbiome dysbiosis and the beneficial effects of probiotics have also been seen in cases of bacterial pneumonia [157], ventilator-associated pneumonia [158], and other upper and lower respiratory tract infections [159]. A cross-sectional study of 30 patients with COVID-19 found significantly reduced intestinal bacterial diversity compared to healthy controls [52], while there was an increase in various bacterial species, including *Morganella morganii, Streptococcus infantis,* and *Collinsella aerofaciens* [53]. These and other studies demonstrate the link between intestinal health and respiratory illness; however, more clinical and randomized controlled trials are needed to better understand the influence of probiotics and their mechanisms of action on SARS-CoV-2. Currently, several randomized studies that started in 2020 and 2021 are underway in the United States, Spain, Italy, and China, examining the effect of specific probiotic strains on the severity of COVID-19, that may directly impact patient care in this pandemic [54,160]. Although probiotics are generally safe with mild side effects, some adverse effects have been reported in patients with underlying health issues. Thus, further research is warranted not only on the efficacy of probiotics as a treatment option for COVID-19 but its safety profile [161,162].

## 11. Discussion

As of December 2021, the SARS-CoV-2 virus has caused approximately 803,000 deaths in the United States. Although the vaccines and newly developed therapeutics helped curb the mortality caused by the virus significantly, the threat of contracting and spreading COVID-19 is far from over. With the potential progression of COVID-19 becoming endemic in many areas of the world, preventing infection and reducing disease severity is still of critical importance. Numerous studies have examined the role of various diets, dietary supplements, and medications in preventing, managing, and treating COVID-19. Our primary aim was to synthesize this body of evidence, identify gaps, and reveal areas for further research. This review systematically presented the current knowledge on diet (Mediterranean versus Western); vitamins D, C, E, and A supplementation; glutathione supplementation; zinc supplementation; traditional Chinese medicine; and probiotics supplementation. The mechanisms of actions of these diets and supplements, the hypotheses on how they might reduce COVID-19 symptoms, and the findings from various clinical studies have also been reviewed. 

Current research indicates that nutritional status may play a valuable role in both the prevention and management of COVID-19 infection. This is largely influenced by diet and supplementation, both of which impact the biochemical and cellular processes involved in the pathophysiology of the disease. Overall, compared to the Western diet, the Mediterranean diet has significant anti-inflammatory and antioxidation effects that may reduce respiratory symptoms and mortality. Some supplements, such as vitamins D and C, have well-established potent immunomodulatory and antioxidant effects, including reducing pro-inflammatory cytokine production and protecting against free radical damage. These have translated to decreased symptom duration and severity, although the support for reducing incidence is still inconclusive. Other vitamins and supplements, such as vitamins A and E and probiotics, have been less researched to support their efficacy in COVID-19 prevention and treatment but may affect innate and adaptive immunity, including T cells and natural killer cells. Zinc and glutathione have been found to affect the disease processes, due to their antioxidant properties; their impact on viral gene expression, viral DNA, and protein synthesis; and their role in maintaining the respiratory epithelium barrier. Studies have not only shown that supplementation with these vitamins and minerals has improved the disease process in some way, but also demonstrated that deficiency can worsen symptoms or prognosis. Traditional Chinese medicine also may have a role in adjuvant COVID-19 therapy, as it has demonstrated many of these same effects for multiple stages of the disease. Overall, while there are some inconclusive and contradictory data, these findings may indicate the potential value of a well-balanced anti-inflammatory diet and supplementation in fighting COVID-19.

Our review synthesizes an extensive portion of the current observational, clinical, and experimental studies published in the international literature up to 2021, complementing other existing reviews. To our knowledge, ours is the first to systemically review the evidence on traditional Chinese medicine and probiotics, along with various vitamins and minerals. However, there are several limitations worth noting. There has likely been new evidence that has surfaced since our search, due to the rapidly evolving nature of the pandemic. Additionally, many of the studies included were observational or cross-sectional studies, limiting the ability for researchers to conclude causation. Among the experimental trials, including blinded randomized studies and open-label trials, protocols were varied, and a vitamin or mineral was often administered in conjunction with another, and a high degree of heterogeneity in their measurements was often present. This further complicates the ability to draw robust conclusions on the effects of one vitamin. The lack of standardized definitions as to what constitutes mild, moderate, or severe disease and the lack of information on patients’ nutritional status, as well as the heterogenicity of the studies, complicate the ability to compare the efficacy of a vitamin or mineral across studies. An important question to raise is whether a nutrition supplement has a prophylactic or therapeutic action on COVID-19 severity, and, if so, what are the best modalities of their administration (dosage and frequency)? Do they act by targeting SARS-CoV-2 or by boosting the host immune system? These questions require well-designed control studies to determine whether supplementation offers a clinical advantage. Currently, there is a lack of studies investigating the potential prophylactic and therapeutic effects of supplements in patients with different co-morbidities. Furthermore, several studies have caused considerable debate and have been criticized for bias and the small sample size used, as well as for failing to consider other cofounding factors, such as nutrient deficiencies, nutritional status, and dietary habits; preexisting conditions and existing treatments; and numerous risk factors, such as sex, ethnicity, lifestyle, and patients’ history, to name a few (for review, see Reference [163]). It is therefore imperative that clinical trials are designed keeping in mind such factors in order to draw meaningful conclusions and generalize the results. Future systematic reviews may apply more stringent inclusion and exclusion criteria to assess the validity of the current literature. Furthermore, the power of some of the studies used in our review is limited due to the sample size of the research and requires caution when discussing the contribution of nutrients in prevention or treatment of COVID-19. With new randomized controlled trials on the horizon, these studies may further elucidate the relationships between diet and COVID-19 and enable public health professionals to provide evidence-based dietary and nutritional recommendations.

## 12. Conclusions

Nutrition, mineral, and vitamin supplementation has received significant interest during the COVID-19 pandemic. In as much as the role of several trace minerals and vitamins in exerting anti-inflammatory effects and boosting immune defenses against infections have long been demonstrated, processes that also underly SARS-CoV-2 infections, demonstrating their role in mitigating COVID-19 clinical manifestations, have proved to be more challenging. Although the current literature suggests possible associations between diet, nutrient supplementation, and COVID-19, the results are, by and large, mixed and inconsistent. Vitamins C and D and zinc did not reduce overall COVID-19 mortality; however, vitamin D was associated with a lower intubation rate and shorter hospital stay. Similarly, some traditional medicine therapies have been shown to improve markers of inflammation, improve disease symptoms, and reduce recovery time in COVID-19; however, there is much more to learn about the causal effects of particular nutrients and their mechanisms of action on COVID-19 clinical outcomes. Notwithstanding the important issues regarding the limitations of the current studies that call for caution when interpreting and generalizing the results and help make practical clinical recommendations in COVID-19 pathology, optimal nutrition status remains an effective strategy for supporting and preserving a strong immune response that is ready to protect against infections. 

## Figures and Tables

**Figure 1 nutrients-14-01909-f001:**
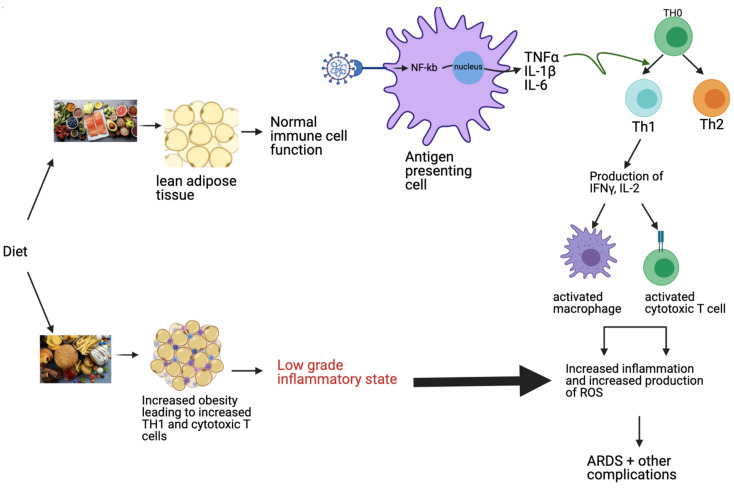
Proposed modulation of the immune pathway by diet. The Western diet, consisting of high fat, refined sugar, and saturated carbohydrates, leads to increased obesity, as well as an increase in TH1 and cytotoxic T cells, resulting in a low-grade inflammatory state. This inflammatory pathway is exacerbated in COVID-19 through the virus’s ability to increase pro-inflammatory cytokines, resulting in ARDS and other morbid complications. In contrast, the Mediterranean diet, consisting of fruits, vegetables, nuts, and fish maintains lean adipose tissue, contributing to a balanced immune system and absent or improved inflammatory state. Nuclear factor kappa B (NF-kb), tumor necrosis factor alpha (TNF-α), interleukin-1 beta (IL-1β); interleukin-6 (IL-6); interferon gamma (IFNγ); interleukin-2 (IL-2); reactive oxygen species (ROS); acute respiratory distress syndrome (ARDS).

**Table 1 nutrients-14-01909-t001:** Effects of nutrient supplementation on COVID-19.

Reference	Country of Study	Design	Study Population, Sample Size	Micronutrient (Dosage)	Effects
Vitamin A					
[18]	Egypt	PR, CT, RD	In progress (Goal: Total *n* = 10,000)	Treatment: 13-cis retinoic acid and standard therapy (4 arms with varying doses) Control: standard therapy	In progress
[19]	USA	PR, OB, CS, CC	*n* = 40 hospitalized patients currently infected with SARS-CoV-2 Case-control *n* = 47 convalescent patients previously infected with SARS-CoV-2	None	S↓ vitamin A levels in hospitalized vs. convalescent patients, and in critically ill vs. moderately ill patients Reduced vitamin A plasma levels correlated with S↑ CRP, LDH, and risk of ARDS, as well as S↓ TLC
Vitamin D					
[20]	Iran	CS, OB	*n* = 235	25OHD	25OHD levels of at least 30 ng/mL were associated with a S↓ in the severity of clinical outcomes related to a COVID-19 infection
[21]	Germany	PR, OB	*n* = 185	25OHD	S↓ serum 25OHD levels in inpatient subgroup vs. outpatient subgroup↑ Rates of hospitalization, IL-6 levels, and intensive oxygen therapy in vitamin-D-deficient patients
[22]	USA	RT, OB, CH	*n* = 489	Vitamin D	Likely deficient vitamin D status associated with S↑ RR of COVID-19 test result
[23]	USA	RT, OB	*n*- = 191,779	25OHD	Higher circulating levels of 25OHD were associated with lower SARS-CoV-2 positivity rate
[24]	United Kingdom	RT, OB	*n* = 449	25OHD	No association between 25OHD and COVID-19 infection
[25]	France	QE	Total *n* = 77 Group 1: *n* = 29 Group 2: *n* = 16 Group 3: *n* = 32	Group 1: regular vitamin D3 supplementation over preceding year (50 K IU, 80 K IU, or 100 K IU) Group 2: vitamin D3 supplementation after diagnosis (80 K IU) Group 3: none	S↓ severe disease and 14-day mortality in Group 1 vs. Group 3 No outcome differences between Groups 1 and 2
[26]	Singapore	OB, CO	*n* = 43	1000 IU/day vitamin D3, 150 mg/d oral magnesium, 500 mcg/d vitamin B12	S↓ clinical deterioration requiring oxygen support, intensive care support, or both
[27]	Brazil	DB, RD, PC, CT	Total *n* = 240 Treatment *n* = 120 Control *n* = 120	Treatment: 200,000 IU of vitamin D3 Control: 10 mL peanut oil solution	S↑ mean serum 25OHD No difference in median length of stay, in-hospital mortality, admission to ICU, and need for mechanical ventilation
[28]	Turkey	Stage 1: RT Stage 2: PR, CT, RD, HC	RT *n* = 162 PR total *n* = 233 Treatment *n* = 210 Control *n* = 23	Vitamin D (100 K IU or 5 K IU for 14 days depending on serum levels of 25OHD)	S↓ 8-day hospital stays; risk of hospitalization
Vitamins C and E					
[29]	Iran	RD, CT, CC	Total *n* = 60 Treatment *n* = 30 Control *n* = 30	Treatment: HD IVC (6 g daily) and Lopinavir/ritonavir and hydroxychloroquine Control: lopinavir/ritonavir and hydroxychloroquine	S↓ mean body temperature and median length hospitalization S↑ peripheral capillary oxygen saturation No S difference in length of ICU stay and mortality
[30]	India	MA	*n* = 6 RCTs	Vitamin C	No change in mortality, length of ICU stay, length of hospital stay, or need for invasive mechanical ventilation
[31]	Iran	RD, CT, DB	Total *n* = 100 critically ill patients with COVID-19 Treatment *n* = 31 Control *n* = 69	Vitamin C	S↓ serum potassium S↑ mean survival duration associated with amount of intake
[32]	Iran	CT, RD	Total *n* = 44 Treatment *n* = 18 Control *n* = 26	Treatment: IV vitamin C (2 g every 6 h for 5 days) and standard treatmentControl: standard treatment	S↑ oxygen saturation S↓ rate of lung involvement and respiratory rate
[33]	China	RD, CT, PC	*n* = 56 critically ill patients	Treatment: vitamin C (12 g/50 mL every 12 h for 7 days)Placebo: water (12 g/50 mL every 12 h for 7 days)	S↑ PaO_2_/FiO_2_, and S↓ IL-6 No difference in invasive mechanical ventilation-free days in 28 days, 28-day mortality
[34]	USA	RT	*n* = 102	Vitamin C and zinc	No impact on overall survival
[35]	USA	RT	*n* = 901	Vitamin C	S↓ mortality
[36]	China	RT	Total *n* = 12 *n* = 6, severe disease *n* = 6, critical disease	Vitamin C (162.7 (71.1–328.6) mg/kg/day for severe patients; 178.6 (133.3–350.6) mg/kg/day for critical patients	S↓ CRP, body temperature, D-dimer, and LDH S↑ TLC
Glutathione and NAC					
[37]	USA	CT, RD, PC	Expected *n* = 64	Treatment: glycine, N-acetylcysteine Placebo: alanine	In progress
[38]	USA	CT	Expected *n* = 48	N-acetylcysteine	In progress
Zinc					
[39]	Brazil	OB, CS	*n* = 269	Serum zinc	Association of low zinc levels and severe ARDS
[40]	Spain	OB, CH	*n* = 249	Serum zinc	Serum zinc levels <50 ug/dL at admission correlated with worse clinical presentation, longer recovery, and higher mortality
[41]	Saudi Arabia	PC, CT, DB, PR, RD	Expected *n* = 40	Treatment: vitamin A 1500 mcg, vitamin C 250 mg, vitamin E 90 mg, selenium 15 μg, and zinc 7.5 mg Placebo: cellulose-containing capsule	In progress
[42]	Saudi Arabia	CT	Expected *n* = 60	Supplement containing quercetin (500 mg), bromelain (500 mg), zinc (50 mg), and vitamin C (1000 mg)	In progress
[43]	Australia	DB, RD, CT, PC	Expected total *n* = 160 *n* = 60 hospitalized patients *n* = 100 ventilated patients	Treatment: zinc 0.5 mg/kg/day Placebo: 250 mL normal saline	In progress
[44]	USA	RD, CT	Total *n* = 214 Arm 1 *n* = 58 Arm 2 *n* = 48 Arm 3 *n* = 58 Arm 4 *n* = 50	Arm 1: zinc gluconate (50 mg) Arm 2: ascorbic acid (8000 mg) Arm 3: zinc gluconate and ascorbic acid Arm 4: standard of care	No significant difference in days required to reach 50% reduction in symptoms among the 4 study groups
Omega-3 Fatty Acids					
[45]	Iran	RD, CT, DB	Total *n* = 101 Treatment *n* = 28 Control *n* = 73	Treatment: omega-3 capsule (1000 mg) containing 400 mg EPAs and 200 mg DHAs for 14 days, and high-protein formula (30 kcal/kg/d) Control: high-protein formula (30 kcal/kg/d)	S↑ 1-month survival rate S↓ BUN, Cr
Traditional Chinese Medicine					
[46]	China	RD, CT	Total *n* = 80 Treatment *n* = 44 Control *n* = 36	Treatment: Jinhua Qinggan granules within 24 h of admission Control: did not take Jinhua Qinggan granules or took the granules for less than 2 d	S↑ viral clearance rate, arterial pH S↓ recovery time indicated by chest CT and mean GCS score No adverse reactions
[47]	China	PR, RD, CT	Total *n* = 284 Treatment *n* = 142 Control *n* = 142	Treatment: usual treatment and Lianhuaqingwen capsule Control: usual treatment	S↑ recovery rate, rate of improvement in chest CT manifestations and clinical cure S↓ median time to symptom recovery, time to recovery of fever, fatigue, and coughing No adverse reactions
[48]	China	RT, CT	Total *n* = 63 Treatment *n* = 37 WM *n* = 26	Treatment: Qingfei Paidu decoction with WM Control: WM	S↑ WBC, TLC, and GOT in both groups S↓ LDH, TLC, BUN, CK, CK-MD, and CRP levels in the treatment group only
[49]	China	RD, DB, CT	Total *n* = 57 *n* = 29 treatment group *n* = 28 control group	Treatment: Xuebijing and routine medication Control: saline and routine medication	S↓ IL-6, IL-8, TNF-a, CRF, rate of mechanical ventilation, septic shock, duration of symptom improvement, length of ICU stay, and proportion of patients who became critically ill S↑ lymphocyte levels
[50]	China	RD, CT	*n* = 60	Treatment: Xuebijing 50 mL or Xuebijing 100 mL BID Control: routine treatment	S↑ WBC count, S↓ CRP, ESR in Xuebijing 100 mL group compared to Xuebijing 50 mL and control groups S↓ CRP and ESR in Xuebijing 50 mL group compared to control group
[51]	China, USA	MA	*n* = 11 7 RCTs, 4 CC	Treatment: Chinese medicine compound drugs (not all studies provided specific drugs utilized) Control: Lopinavir, Ribavirin, Arbidol, and others; nutrition support; and oxygen inhalation	S↑ cure rate and overall response; S↓ hospital stay, illness severity, duration of fever, cough, expectoration, fatigue, chest tightness, and anorexia. No significant difference in adverse drug reactions of nausea, vomiting, diarrhea, liver damage, and reduced RBC count
Probiotics					
[52]	China	CS, OB, CC	Total *n* = 84 *n* = 30 COVID-19 patients *n* = 24 H1N1 patients *n* = 30 HC	None	S↓ bacterial diversity, ↓ relative abundance of beneficial symbionts, and S↑ relative abundance of opportunistic pathogens
[53]	Hong Kong, China	PR, OB	*n* = 15 adults	None	S↑ in *Morganella morganii, Streptococcus infantis,* and *Collinsella aerofaciens* in fecal samples of high-infectivity patients
[54]	USA	RD, CT, DB	Total *n* = 1132	Treatment: *Lactobacillus rhamnosus GG,* 10 billion CFU Placebo: 325 mg microcrystalline cellulose	In progress

Key: MA, meta-analyses; PR, prospective; RT, retrospective; RD, randomized; CT, clinical trial; DB, double-blind; PC, placebo-controlled; OB observational; CS, cross-sectional; CC, case-control; HC healthy controls; QE, quasi-experimental; CFU colony-forming units; S significant; IS insignificant; BID, twice a day; ↑ increase; ↓ decrease; RBC, red blood cell; WBC, white blood cell; CRP, C-reactive protein; ESR, erythrocyte sedimentation rate; IL, interleukin; TNF, tumor necrosis factor; ICU, intensive care unit; CT, chest tomography; PaO_2_/FiO_2_, ratio of arterial oxygen partial pressure to fractional inspired oxygen; WM, Western medicine; LDH, lactate dehydrogenase; TLC, total lymphocyte count; GOT, glutamic-oxaloacetic transaminase; CK, creatine kinase; CK-MB, creatine kinase myocardial band; BUN, blood urea nitrogen; Cr, creatinine; ARDS, acute respiratory distress syndrome; d, day; h, hour; 25OHD, 25 hydroxyvitamin D; RR, relative risk; GCS, Glasgow coma scale; EPA, eicosapentaenoic acid; DHA, docosahexaenoic acid.

## Data Availability

Not applicable.

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
