# Peer review of "Do Diet and Dietary Supplements Mitigate Clinical Outcomes in COVID-19?"

_nutrients, 2022, doi:10.3390/nu14091909_

Round 1

Reviewer 1 Report

The review is highly speculative and operates at many points with assumptions instead of clear studies. In many cases, the claimed evidence refers to other (viral) diseases than COVID-19 and the authors operated with analogy conclusions. It is not clear whether the respective COVID-19 studies were actually randomised and controlled, whether the respective treatment was standardised or whether it was only part of a comprehensive multimodal treatment or a claimed or actual diet was administered. Furthermore, it is not clear what the study size of the trials and their design was.

As an example of the authors' superficial analysis, a continued section is quoted (lines 73-87):
"Most importantly, previous studies showed that a Mediterranean diet can have a positive impact on symptoms associated with rhinitis, asthma and overall respiratory function, reducing oxidative stress in asthmatic children exposed to air pollutants, suggesting a possible protective mechanism of Mediterranean diet against COVID-19 [8]. Considering that inflammation is the hallmark condition of COVID-19, the Mediterranean diet has been recommended not only for overall good health maintenance but also as preventative measures in the fight against inflammatory conditions, including COVID-19. In one ecological study across 17 regions in Spain with Mediterranean diet adherence, there was a negative asso-81 ciation between Mediterranean diet and COVID-19 cases and related deaths. This study further examined 23 countries reporting similar findings of negative association with Mediterranean diet and COVID-19 although more work is needed to establish a causative association [9]. 
In contrast to the Mediterranean diet, the Western diet has been associated with increased inflammatory processes contributing to increased susceptibility to diseases such as CVD, diabetes and COVID-19".

By the way, regarding the "17 regions in Spain with Mediterranean diet adherence",  Spain has just been hit by one of the most severe disease waves in Europe ....

60 lines later, in the section "Gluthatione", the treatment success of NAC was described in 2 (!!!) patients with COVID-19 (Ref. 23), immediately afterwards "successes" in 10 (!!!) COVID-19 patients was reported (Ref. 24). This was followed by a "study" in another 10 (!!!) COVID patients (Ref. 25). All 3 references describe rather a therapy trial than a scientific study. These are the only references to COVID-19 in this section.

Similar procedures are also found in the other sections.

Author Response

Comment: The review is highly speculative and operates at many points with assumptions instead of clear studies.

Response: Thank you for your comments which helped improve our paper. We revised the paper extensively, and more studies have been that included 33 additional references. Specific experimental details, methodology and interpretation have been added to most sections.

Comment: In many cases, the claimed evidence refers to other (viral) diseases than COVID-19 and the authors operated with analogy conclusions. It is not clear whether the respective COVID-19 studies were actually randomised and controlled, whether the respective treatment was standardised or whether it was only part of a comprehensive multimodal treatment or a claimed or actual diet was administered. Furthermore, it is not clear what the study size of the trials and their design was.

Response: We appreciate your comment. We have revised each section and, where appropriate, and if the data was reported in the original article, we included the detailed information pertaining to the experimental design and methodology. For further clarity, we have now included a summary Table (Table 1) for each trace mineral and vitamin presented, that specifies the study design, sample size and principal outcomes.

Comment: As an example of the authors' superficial analysis, a continued section is quoted (lines 73-87):
"Most importantly, previous studies showed that a Mediterranean diet can have a positive impact on symptoms associated with rhinitis, asthma and overall respiratory function, reducing oxidative stress in asthmatic children exposed to air pollutants, suggesting a possible protective mechanism of Mediterranean diet against COVID-19 [8]. Considering that inflammation is the hallmark condition of COVID-19, the Mediterranean diet has been recommended not only for overall good health maintenance but also as preventative measures in the fight against inflammatory conditions, including COVID-19. In one ecological study across 17 regions in Spain with Mediterranean diet adherence, there was a negative asso-81 ciation between Mediterranean diet and COVID-19 cases and related deaths. This study further examined 23 countries reporting similar findings of negative association with Mediterranean diet and COVID-19 although more work is needed to establish a causative association [9]. 
In contrast to the Mediterranean diet, the Western diet has been associated with increased inflammatory processes contributing to increased susceptibility to diseases such as CVD, diabetes and COVID-19".

By the way, regarding the "17 regions in Spain with Mediterranean diet adherence",  Spain has just been hit by one of the most severe disease waves in Europe ....

Response:  We have expanded the discussion on these studies obviating the impact of the diet, the challenges, limitations and complexity when examining the effect of nutrition on disease manifestations. More details of all the studies and in-depth discussion have been added as mentioned above.

Comment: 60 lines later, in the section "Gluthatione", the treatment success of NAC was described in 2 (!!!) patients with COVID-19 (Ref. 23), immediately afterwards "successes" in 10 (!!!) COVID-19 patients was reported (Ref. 24). This was followed by a "study" in another 10 (!!!) COVID patients (Ref. 25). All 3 references describe rather a therapy trial than a scientific study. These are the only references to COVID-19 in this section.

Response: Thank you for your comment. We added six more recent articles on glutathione supplementation and COVID-19 with higher power than the previous studies. On the other hand, its worth noting that studies examining the effect of glutathione and COVID-19 are still lacking and this was noted in the text.

Comment: Similar procedures are also found in the other sections.

Response: We have thoroughly revised the paper and included the experimental and methodological details as well as the basic molecular and metabolic mechanisms of action pertaining to their effects presented.

Reviewer 2 Report

This is an interesting review work on the influence of nutrition on the evolution of covid 19. The review carried out exhaustively and provides important data to take into account.

Author Response

Thank you very much for your positive feedback and appreciation of the topic and the studies presented in this review paper.

Reviewer 3 Report

I found the paper interesting. However, some important literature on this topic was not discussed (e.g. PMID: 33916257)

A table summarizing the potential positive and negative effects of the analyzed interventions might be interesting

Recent evidence (PMID: 25965853, PMID: 33420273) suggest a seasonal dependent effect of vitamin D on health. The authors might discuss the available results on vitamin D (and maybe other supplements) in light of these new concepts.

It has been hypothesized that the home confinement might be associated to nutritional habits modifications and lifestyle modifications (PMID: 32513197). However, available data on a possible increased risk of vitamin D deficiency during the lockdown periods are controversial (PMID: 33924387, PMID: 33925932). I think that it would be important to discuss such issues and suggest practical implications for research and clinics

I would try to separately highlight evidence on children, adults and elderly

Safety issues related to each of identified intervention should be discussed more in detail

I also suggest the authors to consider an important recent systematic review on dietary supplementation and risk of respiratory infections (PMID: 34626488), despite not strictly related to COVID-19

I would better define a conclusion paragraph

Author Response

Thank you for your constructive comments which helped improve our paper.

Comment: I found the paper interesting. However, some important literature on this topic was not discussed (e.g. PMID: 33916257)

Response: Thank you for your comment and for the reference suggested. The paper has been updated to reflect additional information on Vitamin C. This includes additional studies conducted, information on doses and side effects. Thirty-three additional references have been added to the revised paper.

Comment: A table summarizing the potential positive and negative effects of the analyzed interventions might be interesting.

Response:

Thank you for the constructive suggestion. A Table (Table 1) summarizing the effects of nutrient supplementation has now been included.

Comment: Recent evidence (PMID: 25965853, PMID: 33420273) suggest a seasonal dependent effect of vitamin D on health. The authors might discuss the available results on vitamin D (and maybe other supplements) in light of these new concepts.

It has been hypothesized that the home confinement might be associated to nutritional habits modifications and lifestyle modifications (PMID: 32513197). However, available data on a possible increased risk of vitamin D deficiency during the lockdown periods are controversial (PMID: 33924387, PMID: 33925932). I think that it would be important to discuss such issues and suggest practical implications for research and clinics

Response: Thank you for your comment and for the references provided. We have updated the article and discussed the issues of home confinement and increased risk of vitamin D deficiency via  the provided papers (Line 361-380; 445-448;453-456).

Comment: I would try to separately highlight evidence on children, adults and elderly

Response: Thank you for your suggestion. We have tried to review the current literature to determine if there were significant differences between different age groups on the recommended nutritional supplements. However, given the lack of literature on nutritional supplementation for COVID-19 in children, we were unsuccessful in separately highlighting the evidence between the various age groups. We did however, highlighted the effects on specific age groups whenever possible such as in the zinc supplementation section where we reported the effect seen specifically in children.

Comment: Safety issues related to each of identified intervention should be discussed more in detail.

Response: Thank you for your suggestion. We have now included the information related to safety and recommendations for each of the nutrient presented.

Comment: I also suggest the authors to consider an important recent systematic review on dietary supplementation and risk of respiratory infections (PMID: 34626488), despite not strictly related to COVID-19

Reponse: Thank you for your suggestion. The paper has been updated to reflect the addition of the provided paper (Lines 284-287, 374-380).

Comment: I would better define a conclusion paragraph

Response: Thank you for the suggestion. The paper has been revised to now include a conclusion section at the end of the article. 

Reviewer 4 Report

This study was performed on the effect of dietary supplements on Covid-19. The subject of the study is interesting and if its major drawbacks are eliminated, it can be considered for publication.
- The most important drawback of the paper is that since it is not a systematic review and is a narrative, its innovation is questionable. Recently, many review studies have been conducted on the effect of nutrients on the risk of Covid-19 (e.g. https://doi.org/10.1016/j.envres.2020.110053 and https://doi.org/10.1080/13813455.2020.1791188). These paper should be mentioned in the introduction and the strengths of the present study in comparison to these studies should be mentioned.
- The purpose of this study should be clear. The title of the paper is such that the purpose of the study seems to be the effect of nutrients in the prevention of Covid-19, while the text of the article refers to the role of nutrients in the treatment of Covid-19.

- The content of the article needs a major revision. Recently, many clinical trial studies have been performed on Covid-19 patients that the authors did not mention in this paper (e.g., https://doi.org/10.3390/nu12123760, https://doi.org/10.1016/j.jsbmb.2020.105771, and https://dx.doi.org/10.3389%2Ffimmu.2021.717816), while articles on other viral diseases have been considered comprehensively.
Logical order is not seen in the nutrients provided. It is recommended that water-soluble and fat-soluble vitamins and minerals be mentioned first, followed by other dietary components.
Some of the nutrients that have been considered in the treatment of Covid-19 have not been included in this study or have been given very little attention, such as omega-3 fatty acids (e.g. https://doi.org/10.1186/s12967-021-02795-5).

Author Response

Thank you for your positive and constructive comments that helped improve our paper.

Comment: This study was performed on the effect of dietary supplements on Covid-19. The subject of the study is interesting and if its major drawbacks are eliminated, it can be considered for publication.The most important drawback of the paper is that since it is not a systematic review and is a narrative, its innovation is questionable. Recently, many review studies have been conducted on the effect of nutrients on the risk of Covid-19 (e.g. https://doi.org/10.1016/j.envres.2020.110053 and https://doi.org/10.1080/13813455.2020.1791188). These paper should be mentioned in the introduction and the strengths of the present study in comparison to these studies should be mentioned.

Response: Thank you for your comment. We have updated the paper to acknowledge previous literature published on this topic while highlighting the strengths of our present study (Lines 50-72).

Comment: The purpose of this study should be clear. The title of the paper is such that the purpose of the study seems to be the effect of nutrients in the prevention of Covid-19, while the text of the article refers to the role of nutrients in the treatment of Covid-19.

Response: We appreciate the feedback and have revised the title to now state: “Do diet and dietary supplements mitigate clinical outcomes in COVID-19?”

Comment: The content of the article needs a major revision. Recently, many clinical trial studies have been performed on Covid-19 patients that the authors did not mention in this paper (e.g., https://doi.org/10.3390/nu12123760, https://doi.org/10.1016/j.jsbmb.2020.105771, and https://dx.doi.org/10.3389%2Ffimmu.2021.717816), while articles on other viral diseases have been considered comprehensively.

Response: Thank you for the feedback and our apologies for the oversight and not addressing the recent clinical trials. We have now updated all our sections to include the recent clinical trials and other similar studies conducted (Lines 391-393, Lines 355-370, 309-319, 280-284, 190-199, 481-503).

Comment: Logical order is not seen in the nutrients provided. It is recommended that water-soluble and fat-soluble vitamins and minerals be mentioned first, followed by other dietary components.

Response: Thank you for your suggestion. As requested, we have revised the order of the paper to include: vitamin A, vitamin D, vitamin C and E, glutathione, zinc, traditional Chinese medicine and probiotics. 

Comment: Some of the nutrients that have been considered in the treatment of Covid-19 have not been included in this study or have been given very little attention, such as omega-3 fatty acids (e.g. https://doi.org/10.1186/s12967-021-02795-5).

Response: Thank you for the feedback. We have now added a new omega-3 fatty acid section (Section 8).

Round 2

Reviewer 1 Report

The changes made by the authors have substantially improved the paper. Nevertheless, the statements still have to be weakened in some places to take account of the uncertain database:

Page 4, line 166 ff: ...lung health and may possibly be used as an additional, important tools ...

Figure 1, legend: Change to: “Proposed modulation of the immune pathway by diet …”

Page 6, 207/208: concerning the “high safety profile”, It should be noted that retinoic acid has a recognised teratogenic potential, which makes an unqualified recommendation problematic.

Page 11/12, lines 456-472: These are very small studies, with patients on individualized multimodal therapy. I have serious doubts that meaningful conclusions can be drawn about the effect of an additional drug in this small number of patients.

The same remarks are true for the glutathione paragraph (lines 456ff): in studies of that small size observed effects can mean everything or nothing. The heterogenicity prevents to draw any scientific conclusion. This is also acknowledged by the authors who note that “It is worth nothing, however, that additional treatment for these patients included hydroxychloroquine, nitazoxanide, Zithromax, pro-biotics, zinc, vitamin C, beta glucan, curcumin, sulforaphane glucosinolate, alpha lipoic acid for one patient and the other treated with azithromycin, hydroxychloroquine, amoxicillin/clavulanate, zinc, vitamin C, and alpha lipoic acid”. Shorten this paragraph to meaningful studies.

References 85 and 86 are incomplete and literature could not be found. I agree to the authors that related to reference 97 “… the study was small and uncontrolled with variable zinc dosing, indicating a need for larger, standardized clinical trials to properly assess the role of zinc therapy”. My conclusion is to omit the study.

Page 14, second paragraph: Omit lines 559-568 because they are dealing not with COVID-19 patients.

In the next paragraph again, lines 531ff, the results of FOUR patients are extensively discussed.

Page 15, reference 120: 128 critically ill patients (meaning patients on ICU and possibly mechanically ventilated!) were studied and patients on additional omega 3 fatty acids “had a higher one-month survival rate”. The paper itself states: “The intervention group had significantly higher 1-month survival rate compared with the control group (21% vs 3%, P=0.003). About 21% (n=6) of the participants in the intervention group and only about 3% (n=2) of the participants in the control group survived at least for 1 month after the beginning of the study” – The difference was 21% - but the difference was only 4  in 128 critically ill patients with completely individualized standard of care treatment. This is overemphasizing small effects that could be a caused by simple statistical artefact and is neglecting heterogenicity!

Discussion and conclusion: Please, better outline the limitations

Spelling: please write consistently “vitamin D” or “Vitamin D”, always “COVID-1”, throughout the manuscript; line 37: “IFN-g", lines 287-301: “post-oubreak”, “months . were”, “breakout”, and “peripheral oxygen saturation SpO2” instead of “spO2”, line 608: “pH”. Check spelling especially in the changed parts

Author Response

The changes made by the authors have substantially improved the paper. Nevertheless, the statements still have to be weakened in some places to take account of the uncertain database:

Thank you for your positive and constructive comments that helped improved our paper. 

Page 4, line 166 ff: ...lung health and may possibly be used as an additional, importanttools ...

Response: The word “important” has been removed (Line 154).

Figure 1, legend: Change to: “Proposed modulation of the immune pathway by diet …”

Response: The title has been changed, as requested (Line 165).

Page 6, 207/208: concerning the “high safety profile”, It should be noted that retinoic acid has a recognized teratogenic potential, which makes an unqualified recommendation problematic.

Response: We have updated the paper to include vitamin A teratogenic effects (Line 202-203).

Page 11/12, lines 456-472: These are very small studies, with patients on individualized multimodal therapy. I have serious doubts that meaningful conclusions can be drawn about the effect of an additional drug in this small number of patients.

Response: The studies of glutathione involving 2 and 10 cases, respectively, have been removed (Line 455).  

The same remarks are true for the glutathione paragraph (lines 456ff): in studies of that small size observed effects can mean everything or nothing. The heterogenicity prevents to draw any scientific conclusion. This is also acknowledged by the authors who note that “It is worth nothing, however, that additional treatment for these patients included hydroxychloroquine, nitazoxanide, Zithromax, pro-biotics, zinc, vitamin C, beta glucan, curcumin, sulforaphane glucosinolate, alpha lipoic acid for one patient and the other treated with azithromycin, hydroxychloroquine, amoxicillin/clavulanate, zinc, vitamin C, and alpha lipoic acid”. Shorten this paragraph to meaningful studies.

Response: The studies with small sample size have been removed, as requested (Line 450).

References 85 and 86 are incomplete and literature could not be found.

Response: We apologize for the error. References were reformatted, as per journal guidelines for citing website sources (Line 1103-1104).

I agree to the authors that related to reference 97 “… the study was small and uncontrolled with variable zinc dosing, indicating a need for larger, standardized clinical trials to properly assess the role of zinc therapy”. My conclusion is to omit the study.

Response: This study was omitted (Line 557).

Page 14, second paragraph: Omit lines 559-568 because they are dealing not with COVID-19 patients.

Response: The suggested lines were omitted (Line 353).

In the next paragraph again, lines 531ff, the results of FOUR patients are extensively discussed.

Response: The study was omitted.

Page 15, reference 120: 128 critically ill patients (meaning patients on ICU and possibly mechanically ventilated!) were studied and patients on additional omega 3 fatty acids “had a higher one-month survival rate”. The paper itself states: “The intervention group had significantly higher 1-month survival rate compared with the control group (21% vs 3%, P=0.003). About 21% (n=6) of the participants in the intervention group and only about 3% (n=2) of the participants in the control group survived at least for 1 month after the beginning of the study” – The difference was 21% - but the difference was only 4  in 128 critically ill patients with completely individualized standard of care treatment. This is overemphasizing small effects that could be a caused by simple statistical artefact and is neglecting heterogenicity!

Response: We have expanded the discussion/conclusion section, including the limitations and added three more references (Line 620-638).

Spelling: please write consistently “vitamin D” or “Vitamin D”, always “COVID-1”, throughout the manuscript; line 37: “IFN-g", lines 287-301: “post-oubreak”, “months . were”, “breakout”, and “peripheral oxygen saturation SpO2” instead of “spO2”, line 608: “pH”. Check spelling especially in the changed parts.

Response: All spelling and grammatical errors have been corrected.

Reviewer 4 Report

The authors have made the necessary corrections to the paper.

Author Response

Thank you for your positive comments.